# Prevalence of Malnutrition in Patients with Parkinson’s Disease: A Systematic Review

**DOI:** 10.3390/nu14235194

**Published:** 2022-12-06

**Authors:** Karolina Wioletta Kacprzyk, Magdalena Milewska, Alicja Zarnowska, Mariusz Panczyk, Gabriela Rokicka, Dorota Szostak-Wegierek

**Affiliations:** 1Department of Clinical Dietetics, Faculty of Health Sciences, Medical University of Warsaw, 00-581 Warsaw, Poland; 2Department of Education and Research in Health Sciences, Faculty of Health Science, Medical University of Warsaw, 00-581 Warsaw, Poland

**Keywords:** Parkinson’s disease, malnutrition, prevalence, systematic review

## Abstract

Objectives: This systematic review analyzed the prevalence of malnutrition in patients with Parkinson’s Disease. Study design: a systematic review. Method: Four databases—Cochrane, PubMed, Embase and Web of Science—were searched from October 2021 to June 2022 by two independent researchers. The inclusion criteria were as follows: patients above 18 years old with confirmed Parkinson’s Disease, performed screening nutritional assessment, cohort studies, case-control studies, and cross-sectional studies. Patients without Parkinson’s Disease and with other parkinsonian syndromes were excluded. Results: 49 studies were included in this systematic review. Patients ranged in age from 20 to 96 years. There were 5613 subjects included. According to Mini Nutritional Assessment, 23.9% (*n* = 634) participants were at risk of malnutrition and 11.1% (*n* = 294) were malnourished. According to BMI score, most patients were either obese or overweight. Conclusions: the prevalence of malnutrition or risk of malnutrition in the study group was significant. Therefore, more specific and detailed studies on the prevalence of malnutrition in patients with Parkinson’s Disease are needed.

## 1. Introduction

Parkinson’s disease is described as a progressive disorder characterized by degeneration of the dopaminergic neurons of the substantia nigra in the midbrain [1] and the presence of alpha-synuclein-positive cytoplasmic inclusions called Lewy bodies [2]. It was first described by British doctor James Parkinson in the “Essay on Shaking Palsy” in 1817. [3] Patients experience progressing decreases in motor and cognitive functions and higher mortality [1]. The most common symptoms are bradykinesia, rigidity, postural reflex impairment, and resting tremor [1,3]. Some psychiatric symptoms, such as anxiety, dementia and depression, can be noted in patients with Parkinson’s disease [3], as well as sleep disorders, especially insomnia and REM behavioral disorder [2]. Other noticeable symptoms are fatigue, constipation, hypotension, cramps, and seborrheic dermatitis [3]. 

The diagnosis is especially hard in the early stages of the disease because its symptoms are common to other conditions such as vascular parkinsonism, essential tremor, or progressive supranuclear palsy [1]. The diagnostics are based on clinical criteria, such as parkinsonism and no history of drugs, toxins, or infections, also excluding the signs of other neurological damage [3]. Most PD cases are caused sporadically, and only a small number of patients carry disease-causing genetic mutations [2].

There are two groups of substances that can be used in the treatment of Parkinson’s disease: levodopa and monoamine oxidase-B inhibitors. Levodopa is an agonist of dopamine and is administered with carbidopa. It inhibits the metabolism of levodopa and allows the therapeutic concentration of the drug to enter the brain [1]. It is the most effective drug used in Parkinson’s disease treatment. Its chemical structure allows it to compete with neutral amino acids for intestinal absorption. Due to that, levodopa should be administered at least 30 min before each meal. This facilitates the avoidance of drug interaction with dietary protein. Side effects of antiparkinsonian drugs are often reported by patients as symptoms contributing to an increased risk of weight loss. Among the most frequently mentioned side effects are: abdominal pain, vomiting, dyspepsia, dry mouth, diarrhea or constipation, and other gastrointestinal disorders [4,5,6].

One of the conditions which leads to an increased risk of malnutrition in PD patients is oropharyngeal dysphagia. It is estimated that around 80% of patients suffer from this kind of issue in the early stages of the disease. The incidence increases to 95% in later stages. Delayed oropharyngeal transition time and reduced muscle strength are most common in this group of patients. Due to dehydration, malnutrition, possible silent penetration, and aspiration of consumed foods to the lungs, the mortality rate increases significantly [7]. Swallowing dysfunctions in PD patients are diagnosed using the VFSS and FEES tests. A multidisciplinary approach is necessary for the proper management of dysphagia in PD patients [8]. 

Patients with Parkinson’s disease are at high risk of weight loss and malnutrition. Weight loss may be present at diagnosis and is associated with increased energy expenditure due to tremors and rigidity. It leads to an increase in the daily dose of levodopa, which results in the worsening of dyskinesias. Weight gain can be noticed as well. It usually occurs because of dopaminergic treatment or deep brain stimulation (DBS). This should be taken into consideration, since it usually results in a fat mass gain, mostly in the abdominal area, and leads to an increased risk of metabolic syndrome [9].

Due to the many negative aspects of Parkinson’s disease (such as the aforementioned unintentional changes in body weight, swallowing difficulties, drug side effects, and more), there is a need for regular evaluation of nutritional status among this group of patients. 

This study aimed to review the nutritional status of patients with Parkinson’s disease and to determine the prevalence of malnutrition in the mentioned group—and to assess the need for nutritional status screening.

## 2. Materials and Methods

This study was written based on the Preferred Reporting Items for Systematic Reviews and Meta-analyses (PRISMA) guidelines and was not registered in PROSPERO.

### 2.1. Literature Search and Strategy

A literature search was conducted via four databases: Cochrane, PubMed, Embase and Web of Science. We included papers that were published between 2000 and 2020. The search was conducted using Medical Science Heading (MeSH) terms (Table 1). Relevant articles were accessed in full text and checked for eligibility criteria. The analysis began in October 2021 and ended in June 2022. Figure 1 presents the methodology if the search.

### 2.2. Inclusion and Exclusion Criteria

Studies included in our review had to include a nutritional status assessment method: use of a specific questionnaire or measurement of the BMI. Some studies did not involve the exact topic (the prevalence of malnutrition in PD patients), but had to include participants with this condition. 

Studies were included based on the following criteria:adult patients above the age of 18,patients diagnosed with Parkinson’s disease, andscreening nutritional assessment,Exclusion criteria were:animal research,wrong study publication (such as posters or conference abstracts),studies on cells in vitro,patients diagnosed with other parkinsonian syndromes, andunavailability of the abstract and full text.

We also excluded studies if there was no response from the authors after 14 days from contact. Language was not considered an exclusion criterion in our study. 

We contacted the corresponding authors via correspondence e-mail or other e-mail addresses found on the internet. After 14 days without reply, we either excluded the study or marked lacking information as “nd” (no data). 

### 2.3. Eligibility Criteria 

The PICOS framework was used to specify qualification requirements. The population eligible for inclusion was only the adult population (P). Screening tools that were taken into consideration were: MNA, MNA-SF, SGA, PG-SGA, MUST, MEONF-II, SCREEN-II and Body Mass Index (I). There was no comparison (C), and the outcome of interest (O) was the prevalence of malnutrition and risk of malnutrition. As for the methodology of the study, the cross-sectional studies, cohort studies and case-control studies were acceptable. Posters and clinical cases were excluded. The PICOS structure criteria for inclusion are shown in Table 2.

### 2.4. Data Extraction

All eligible abstracts were transferred to Rayyan, where two independent reviewers (KK and AZ) conducted an analysis based on title and abstract. All studies that met the inclusion criteria were included in further searches, while those that met the exclusion criteria were excluded. The uncertainties were resolved by two other researchers (MM and GR). 

If the data were not clear, full text was reviewed, based on which the study was either excluded or included.

Further searches consisted of the analysis of full text of papers included in the abstract analysis. All articles were consulted between all participating reviewers (KK, AZ, GR and MM) and all discrepancies were resolved. 

The data were maintained in a Microsoft Excel spreadsheet. The following data were collected: author, year of the study, country, study type, study sample, Hoehn & Yahr stage, UPDRS total score, age (minimum, maximum, standard deviation), sex, assessment method, scores of each questionnaire (mean, minimum, maximum, standard deviation), and BMI score. 

In case of absence of data needed for study inclusion, authors were contacted via e-mail provided for correspondence or other address found on the Internet. Lack of response from the corresponding authors within 14 days resulted in exclusion of the study due to absence of required data. 

### 2.5. Quality Assessment and Statistical Analysis

Quality assessment of the included studies was carried out using the Critical Appraisal tools for use in JBI Systematic Reviews. The means of this assessment was to see how the study assessed the probability of bias in its planning, analysis, and execution. Only scores over 50% were included. Detailed data are provided in Figure 2, Figure 3, Figure 4 and Figure 5.

Results were reported with standard deviation (SD), further combined, and analyzed in MS Excel.

## 3. Results

There were 49 studies included, evaluating patients’ nutritional status using different assessment methods (body mass index, questionnaires: MNA, MNA-SF, SGA, PG-SGA, MUST, MEONF-II and SCREEN-II), resulting in various study designs (cross sectional: 33; case control: 10; cohort studies: 6).

### 3.1. Group Characteristics

There were 5727 patients in all. Studies assessed originated from all continents except Africa and Antarctica; 57.3% of patients were men and 40.9% were women. Most studies were conducted in Asia (24 studies), with the second most in Europe (16 studies). There were 2 studies in Australia, 4 in North America, and 3 in South America. More detailed data are presented in Table 3.

### 3.2. Nutritional Status Assessment

#### 3.2.1. Malnutrition Universal Screening Tool (MUST)

Only two studies [15,29] assessed nutritional status using the Malnutrition Universal Screening Tool (MUST). In this section, there were 343 participants, but the authors of each study showed different results. In the first study [15], 17.2% of patients were at risk of malnutrition and 5% were already malnourished. The second study [29] pointed out that 15.4% of patients were at medium risk of malnutrition, 8,1% were at high risk of malnutrition, and the rest (76.5%) were at low risk of malnutrition. 

There was a problem with comparing the two studies, as one provided more specific data (such as mean score of MUST questionnaire for the group of patients [15]) than the other. Additionally, the authors described nutritional status in different ways. Nonetheless, both studies presented less than 10% of their groups at either high risk of malnutrition or already malnourished, and between 15 and 17% at moderate risk or at risk in general.

More detailed data are provided in the table below (Table 4).

#### 3.2.2. Subjective Global Assessment (SGA) and Patient Generated Subjective Global Assessment (PG-SGA)

In this section, there were only two studies [30,31]. These were conducted by the same authors, and included a total of 250 patients, of which 58.8% were male and 41.2% were female. The median score of both questionnaires was 3. In the first study, 15% of patients were categorized as SGA-B, i.e., at risk of malnutrition or moderate malnutrition [30]. The second study showed similar results, with 15.2% of patients categorized as SGA-B [31]. No participants were assigned to the SGA-C group, which referred to severe malnutrition [30,31].

More specific data are available in Table 5 and Table 6.

#### 3.2.3. SCREEN-II and MEONF-II

Seniors in the Community: Risk Evaluation for Eating and Nutrition (SCREEN-II) and Minimal Eating Observation and Nutrition Form—Version II (MEONF—II) were used in only one study by Lindskov, S.; Sjöberg, K. et. al. [16]. The study provided data on the onset of the study and at a follow up one year later. SCREEN-II results showed that 69.4% were at high risk of malnutrition and 22.6% at moderate risk; these later rose to 75.8% and fell to 21%, respectively. MEONF-II was used to assess the risk of undernutrition; 96.9% of participants were at low risk of undernutrition and 0% were at high risk at the baseline. Follow-up assessment showed a slight drop in undernutrition low risk group (95.4%), while high risk rose to 1.5% [16].

More specific data are provided in the Table (Table 7) below.

#### 3.2.4. Mini Nutritional Assessment (MNA)

Mini Nutritional Assessment is used for assessing the nutritional status of patients. There were 22 studies included in which MNA assessments were conducted [10,11,12,16,17,18,19,21,32,33,34,35,36,37,38,39,40,41,42,43,44,45]. There was a total of 2713 participants, of which 39.2% were female and 60.8% were male. The youngest patient included was 20 years old [17], while the oldest was 92 years old [10]. The highest reported number of malnourished patients was 39.2% [10]; at risk of malnutrition was 59% [32], and the lowest were 0% and 14%, respectively [21]. A significant number of studies (6 out of 22) did not report on the prevalence of malnutrition or risk of malnutrition [16,17,19,35,37,44]. The summarized results of MNA questionnaire are shown in Table 8. One study reported the nutritional risk levels as low, moderate, and high, respectively, on each stage of the study: baseline and follow-up [16]. The nutritional status report of this study [16] is not included in the table below summarizing nutritional status of patients using the MNA questionnaire (Table 9). It can be found in Table 10.

##### MNA-SF

Three studies reported the nutritional status of PD patients using the MNA Short Form (MNA-SF) [22,32,46]. In this section, 147 patients were included, consisting of 54.4% male participants and 45.6% female participants. The maximum age was 94 years [46] and minimum was 65 [22]. One study reported an inclusion criteria minimal age of 45 but did not state the exact lowest age of its subjects [32]. Only one study reported on the percentage of patients at risk of malnutrition and malnourished patients: 14.8% and 5.6%, respectively [22]. One study reported only on the number of those at risk of malnutrition, and the last did not contain that information [32].

The number of studies using MNA-SF as assessment tool were insufficient to compare the results in adequate and valuable way. 

The combined results are shown in Table 11.

##### Body Mass Index (BMI)

We took into consideration 49 studies [10,11,12,13,14,15,16,17,18,19,20,21,22,23,24,25,26,27,28,29,30,31,32,33,34,35,36,37,38,39,40,41,42,43,44,45,46,47,48,49,50,51,52,53,54,55,56,57,58] in which BMI was assessed, resulting in 5727 patients, 57.3% of whom were men and 40.9% of whom were women. The lowest reported mean BMI was 17.84 kg/m^2^ [46] and the highest was 30.17 kg/m^2^ [23]. Five studies did not report BMI score despite using it in the process of the study as an assessment tool [11,17,30,34,44]. Wang G. et al. reported the BMI scores for a group of MNA ≤ 23.5 but did not provide the same data for MNIA > 23.5 [40]. Some studies reported the mean BMI score as the median instead of the mean [31,56]. This resulted in difficulties comparing the data of all studies in this section and, along with the absence of data from other studies, made it impossible to assess the mean BMI of the entire group of participants included in our study. 

More detailed data are provided in the table (Table 12) below.

## 4. Discussion

Our systematic review aimed to show the prevalence of malnutrition among patients suffering from Parkinson’s disease. We included 49 studies from 21 countries. We noted that there were disbalances in the results of each questionnaire and method of nutritional status assessment. 

Malnutrition is a common issue experienced by patients suffering from Parkinson’s Disease which results in poor health of the patients. Adequate assessment methods and management strategies are needed, as malnutrition affects these groups excessively. There are many coexisting factors that should be taken into consideration at the onset of the disease and during evaluations, such as probable swallowing disorders. Dysphagia affects a significant number of PD patients, and severe dysphagia appears mostly in advanced stages of Parkinson’s disease. Hoehn and Yahr stages 4 and 5, recent weight loss, sialorrhea, and BMI < 20 kg/m^2^ have been considered to predict dysphagia in the mentioned group [8].

Weight loss is another factor strongly associated with malnutrition in PD patients, as it results in cognitive decline, orthostatic hypotension, and dyskinesia. These, in turn, lead to the intake of a higher dosage of levodopa. Both weight loss and malnutrition are a result of negative energy balance, meaning that the intake is lower than the expenditure of energy [59]. It can also be related to hyposmia, which is a decrease in the capability to sense a smell. Hyposmia is also considered to be one of the earliest non-motor symptoms of PD [60]. 

Energy balance is known as homeostasis and is a process in which the human body adjusts food intake and energy expenditure. In healthy individuals, the input of energy equals the output. In PD patients, both disease-related and treatment-related factors can contribute to disturbed energy balance correlated with dopaminergic deficit. In early stages of Parkinson’s disease, symptoms like sensory disfunction or gastrointestinal disfunction are associated with weight loss due to decreased food consumption. In later stages, when motor symptoms are more common, energy expenditure increases. [59] Resting energy expenditure (REE) is hard to assess. Some authors pointed out that REE was higher in untreated patients than in those treated with L-dopa [61], while some authors suggested that REE remained unchanged between untreated and treated individuals [62]. REE is associated with tremor and rigidity. Those two symptoms are strongly associated with Parkinson’s disease and contribute to increased energy expenditure and weight loss.

Another predictor of weight loss and future risk of malnutrition is gastrointestinal dysfunction, which is common in PD patients and can precede motor defects by years. Parkinson’s Disease is associated with intestinal inflammation and other gastrointestinal abnormalities, such as constipation or dysbiosis [63]. 

Early detection of weight loss can lead to early notice of malnutrition risk, which makes it possible to improve a patient’s quality of life. Nutritional intervention is meant to improve negative aspects of Parkinson’s disease (e.g., managing swallowing difficulties or maintaining proper body weight), resulting in better quality of life and helping to reduce the progression of the disease. A study by Sheard J. et al. showed that malnourished patients experienced lower life quality than well-nourished patients—and that nutritional intervention helped to improve emotional well-being [64]. The authors concluded that early detection of weight loss could help to prevent malnutrition and reduction in quality of patients’ lives. 

Various tools are used to screen the nutritional status of patients. Aside from biochemical factors or recent weight loss, specific questionnaires are currently in use. One of the most common is the Malnutrition Universal Screening Tool (MUST), which was developed to assess patients in all kinds of ambulatory care [65]. It includes body mass index value, weight loss for the past 3–6 months, and anorexia associated with an illness in the past 5 days. It also includes two questions about unintentional weight loss and a decrease in food intake [66]. 

The Seniors in Community: Risk Evaluation for Eating and Nutrition (SCREEN II) questionnaire consists of seventeen questions and assesses the risk of malnutrition based on the number of eaten meals, difficulties in eating such as dysphagia, changes in body mass, and social aspects of food intake in patients, e.g., living alone in their homes [66]. 

Subjective Global Assessment (SGA) is the most commonly used tool for assessing nutritional status. It includes two kinds of information: medical history and physical examination. The first part includes recent weight loss, functional impairment, and changes in dietary intake, whereas the following part brings up loss of subcutaneous fat, oedema, and muscle wasting. Patients are classified as well nourished (SGA A), suspected or moderately malnourished (SGA B), or severely malnourished (SGA C). The downside of this questionnaire is that it does not monitor changes in nutritional status and does not include biochemical values [28].

The MEONF-II screening tool was designed to determine the risk of undernutrition in hospital inpatients. The score ranges from 0 to 8 points. A score of 0–2 is considered a low risk of undernutrition, 3–4 a moderate risk and ≥5 is a high risk of undernutrition [67]. 

The highest sensitivity can be found in the Mini Nutritional Assessment (MNA) questionnaire. It was first introduced to assess the risk of malnutrition in institutionalized geriatric patients and became useful in detecting probable malnutrition in senior patients when other parameters, such as biochemical factors and BMI, are still correct [65]. It includes diverse components, such as: weight loss and depletion in food access in the past 3 months, changes in physical activity, psychological stress, and acute illness in the mentioned period, as well as anthropometric measurements, altered sense of smell and taste, or frailty. The MNA consists of eighteen questions divided into four domains. On the contrary, a shorter version called the MNA-SF consists of six questions and is considered to be as effective as the full sheet. If the MNA-SF score is 11 points or less, then the full version should be applied for proper assessment, and the patient is considered at risk of malnutrition [65].

As can be seen from the characteristics of each questionnaire, some assess the risk of malnutrition among patients (MNA, MUST, and SCREEN-II), and others assess nutritional status in general (SGA). MEONF-II, on the other hand, assesses the risk of undernutrition. 

One of the included studies, written by Yang T., Zhan Z. et al., showed 39.2% of patients assessed for malnutrition using the MNA questionnaire as malnourished and 30% as at risk of malnutrition, whereas 30.8% had good nutritional status [10]. On the contrary, a study by Vikdahl M., Carlsson M. et al. showed no patients with malnutrition, and only 14% of the cohort were deemed at risk of malnutrition [21].

Only one study that used MNA-SF showed the number of patients with malnutrition or at risk. Umay E., Yigman Z. A et al. pointed out that 14.8% of patients were at risk of malnutrition and 5.6% were already malnourished [22].

Two studies included in our review in which authors used the MUST questionnaire showed comparable results. Barichella M., Cereda E. et al. showed 17.2% of patients at risk of malnutrition and 5% already malnourished [15], while Jaafar A.F., Gray W.K. et al. showed 8.1% of patients at high risk of malnutrition and 15.4% of patients at moderate risk of malnutrition [29].

We took into consideration only two studies where authors assessed the patients with SGA and PG-SGA questionnaires. The results showed around 15% of all cohorts, respectively, identified as level B (moderate malnutrition or at risk of malnutrition) and 0% as level C (severely malnourished) [30,31].

There was a problem with comparing results using SCREEN-II and MEONF-II due to an insufficient number of studies that used those tools in the assessment of patients [16].

On the other hand, studies where authors used body mass index as a tool to measure malnutrition did not offer specific data, other than the fact that, in most studies, the mean BMI score showed patients as overweight or obese, with the highest score of 30.17 kg/m^2^ [23] and lowest 17.84 kg/m^2^ [46]. No study that measured only BMI showed the percentage of those malnourished or at risk of malnutrition. The overall results for BMI did not suggest any of the patients to be malnourished or at risk of malnutrition. BMI is used for measuring the weight category, but it is not a reliable way of measuring nutritional status. Thus, BMI is not an adequate assessment tool for assessing patients for malnutrition. 

Many authors opted not to provide information on the exact number of patients at risk of malnutrition or malnourished, only the mean score of the questionnaire for the entire cohort studied [13,14,17,19,20,23,24,25,26,27,28,35,37,44,46,47,48,49,50,51,52,53,54,55,56,57,58].

Comorbidities and community could have affected the results, but those were not taken into consideration while preparing this review. 

Most studies provided information about disease severity in their cohorts. There were two scales used: Hoehn and Yahr scale and the MDS-Unified Parkinson’s Disease Rating Scale (MDS-UPDRS). Some studies only provided the Hoehn and Yahr scale [31,32,37,41,45,51]. Sometimes authors reported the mean score, sometimes the median [17,21,31,38,39,41], and sometimes the percentage of patients on each level of the scale [24,50,54,56]. The same issue applied for the UPDRS scale, where various aspects of the scale were reported. A few studies did not provide any information on the disease severity of the cohort included in their research [11,16,18,19,26,27,30,34,42,46,47,49,52,55]. The authors showed no correlations between the disease severity and the nutritional status of the patients, leading to the conclusion that no connection existed between disease severity and the prevalence of malnutrition.

Biochemical markers are important factors that could provide more detailed insight into the problem of malnutrition. Albumin, prealbumin, total cholesterol, hemoglobin, and total protein could be useful in screening for malnutrition [68], but were not included in our study. Screening assessments alone could be inadequate, pointing to a need for more precise markers, such as biochemical markers; however, due to limitations these needs could not be met. Additionally, weight loss may be hard to observe in overweight or obese patients; due to that more in-depth analysis will be needed. 

Due to the absence of unified diagnostic criteria for malnutrition, ESPEN, ASPEN, PENSA and FELANPE created the Global Leadership Initiative on Malnutrition (GLIM). The first thing needed for diagnosing malnutrition is a screening tool of choice to determine those at risk of malnutrition. The GLIM authors pointed out criteria to look for in patients marked as at risk. Those are as follows: low body mass index, unintentional weight loss, reduced muscle mass, reduced food intake or assimilation, inflammation, or disease burden. The first three were categorized as phenotypic criteria, and the last two as etiologic criteria. For the diagnosis of malnutrition, the authors recommended that a combination of at least one phenotypic and one etiologic criterion is required [69]. Weight loss is easy to assess by a doctor or caretaker of a patient. More attention should be paid to this criterion, as fast detection can be crucial in the proper management of malnutrition.

The GLIM diagnostic criteria for malnutrition should be applied to every patient diagnosed with Parkinson’s Disease, as all criteria mentioned by the authors are common in this group of patients. None of the studies reviewed in our study mentioned the GLIM criteria, and hardly any study was based solely on the problem of malnutrition in PD patients. 

Another important guideline that mentioned the importance of screening for malnutrition in Parkinson’s Disease patients are the ESPEN guidelines for clinical nutrition in neurology. The authors mentioned that malnutrition was underreported in this group of patients and could be found in at least 15% of community-dwelling patients. The other 24% were at medium or high risk of malnutrition. They also pointed out the importance of screening for oropharyngeal dysphagia and gastrointestinal dysmotility. The first recommendation in this group pointed out the need for regular screening of nutritional status and vitamin levels during the disease, with extra attention paid to recent changes in body weight. The guidelines also recommended screening for dysphagia, as early detection can help in proper intervention and prevention of malnutrition [70]. 

The main issue during the writing of this systematic review was the absence of studies investigating the actual problem and prevalence of malnutrition in a group of Parkinson’s disease patients. Most of the data had to be extracted from a variety of studies describing different issues experienced by patients with PD. We also came across a problem when separating the results of only patients with confirmed Parkinson’s Disease. Some studies included larger cohorts with various diseases. We contacted the authors, often with negative results. In the end, studies that lacked necessary information, or in which it was impossible to isolate PD patients from a larger cohort, were excluded from our research.

Currently, only screening tests are available. However, in-depth research is needed to provide adequate, detailed information on the nutritional status of PD patients. Due to the absence of homogeneous studies, it was impossible to conduct results synthesis and meta-analysis. 

Our study assessed the prevalence of malnutrition among patients with Parkinson’s Disease by considering 7 questionnaires (MNA, MNA-SF, SGA, PG-SGA, MUST, MEONF-II and SCREEN-II) and BMI. The outcome of this study showed the need for further and more detailed studies on the prevalence of malnutrition in this group of patients. Additionally, it highlighted the lack of adequate assessment tools. Research in this area should be increased, with more care applied to nutritional assessment, as it can determine the quality of life of a patient. A multidisciplinary approach should be applied, and screening for malnutrition should be a routine test, both in the early stages of the disease and throughout the course of the disease, as the prevalence of malnutrition rises with the progression and severity of the disease.

## 5. Conclusions

This systematic review showed a limited number of papers dedicated to the assessment of nutritional status in patients with Parkinson’s Disease. The studies ranged widely due to the use of several different questionnaires to assess the nutritional status of patients. It was not possible to assess differences in sex, diet, region, economic status, or the severity of Parkinson’s disease. These factors could provide even more insight and help to differentiate more aspects that determine the at-risk level of a patient. Furthermore, it is important to note that many patients tend to be overweight or obese, which should not be ignored while assessing nutritional status. Increased body weight can be seen as a marker of good nutritional status. However, it can be the opposite, and is quite common in patients with Parkinson’s disease. According to available data, the prevalence of malnutrition in patients with Parkinson’s disease is significant, despite many patients displaying excessive body mass. Our results were based on screening assessments. Due to that, further, more detailed research (with greater attention paid to data, methodology, and study design) is needed in this area to provide better insight into the prevalence of malnutrition in patients suffering from Parkinson’s disease. Other, more detailed assessments should be employed which use, for example, biochemical markers or anthropometric measurements. Moreover, it is crucial to properly assess patients’ nutritional status from the onset of the disease in order to apply proper intervention and help to improve patients’ quality of life.

## Figures and Tables

**Figure 1 nutrients-14-05194-f001:**
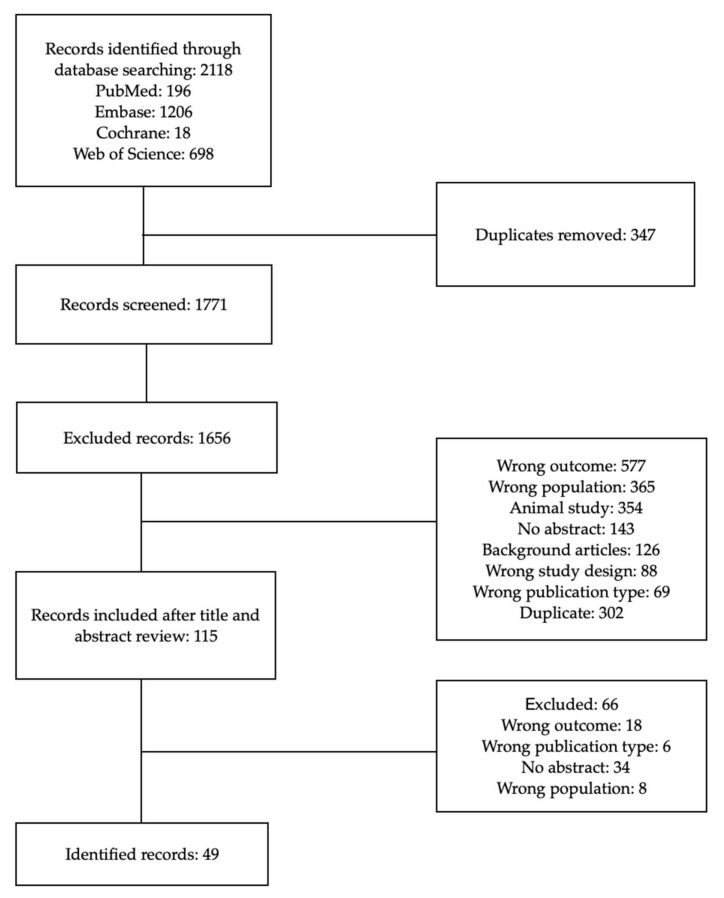
Flowchart.

**Figure 2 nutrients-14-05194-f002:**
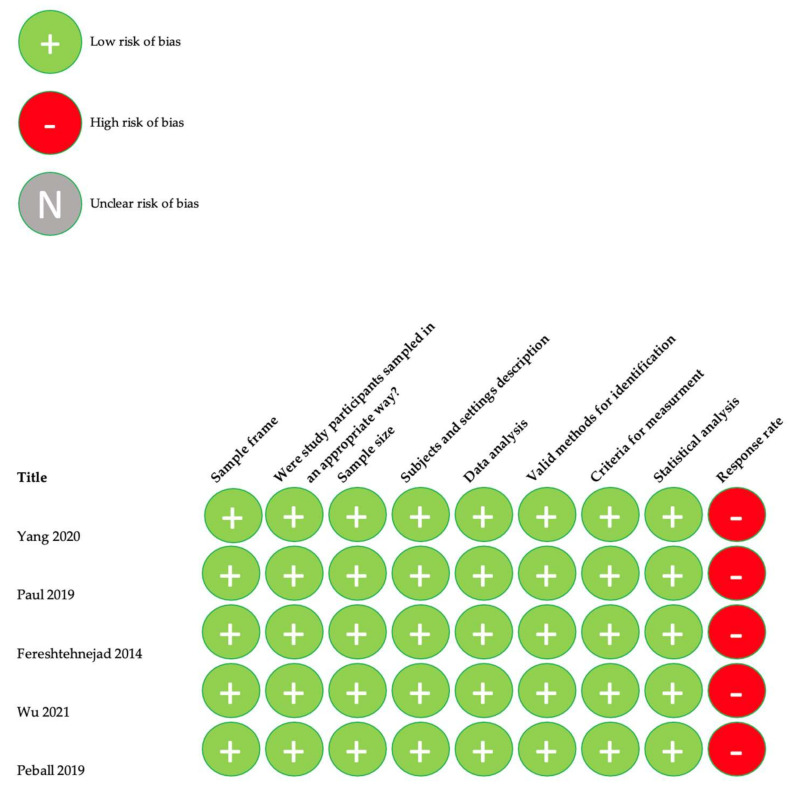
Quality assessment for prevalence studies [10,11,12,13,14].

**Figure 3 nutrients-14-05194-f003:**
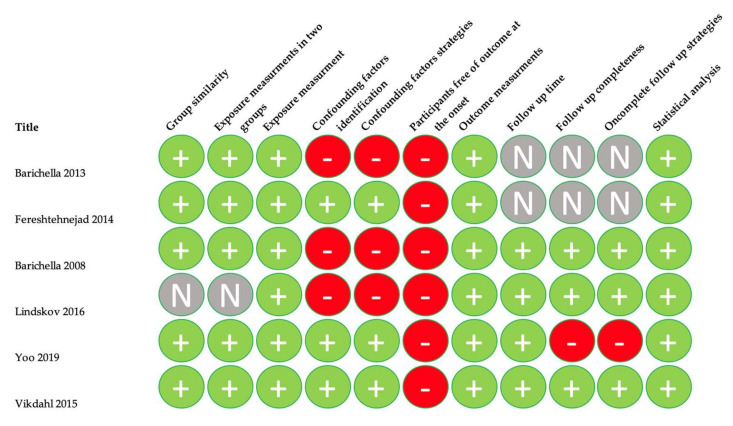
Quality assessment for cohort studies [15,16,17,18,19,20].

**Figure 4 nutrients-14-05194-f004:**
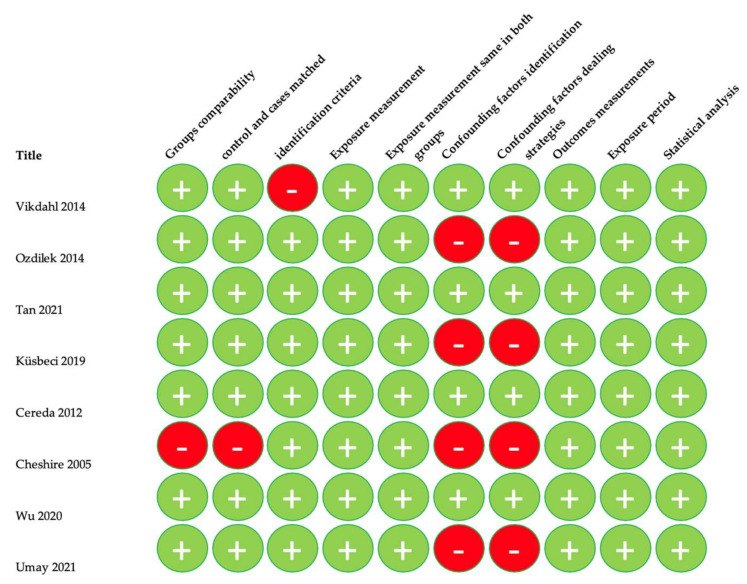
Quality assessment of case control studies [21,22,23,24,25,26,27,28].

**Figure 5 nutrients-14-05194-f005:**
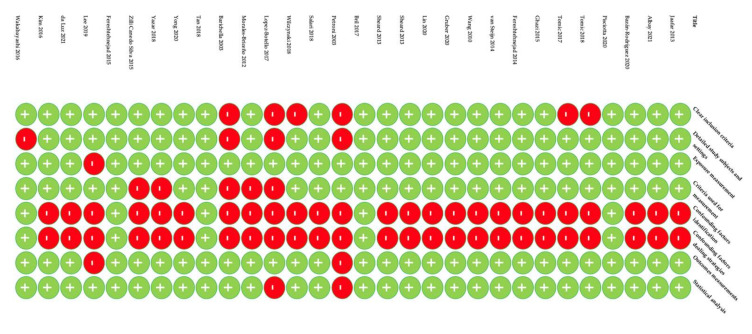
Quality assessment of cross-sectional studies [29,30,31,32,33,34,35,36,37,38,39,40,41,42,43,44,45,46,47,48,49,50,51,52,53,54,55,56,57,58].

**Table 1 nutrients-14-05194-t001:** Medical Science Heading (MeSH) terms.

PubMed	
#1	“Parkinson Disease”[MeSH Terms]
#2	“Weight Loss”[MeSH Terms] OR “Nutritional Status”[MeSH Terms] OR “Cachexia”[MeSH Terms] OR “Malnutrition”[MeSH Terms] OR “Frailty”[MeSH Terms] OR “Sarcopenia”[MeSH Terms] OR “Muscular Atrophy”[MeSH Terms]
#3	“case reports”[Publication Type] OR “comment”[Publication Type] OR “meta analysis”[Publication Type] OR “retracted publication”[Publication Type] OR “retraction of publication”[Publication Type] OR “systematic review”[Publication Type] OR “review”[Publication Type]
(#1 AND #2) NOT #3	(“Parkinson Disease”[MeSH Terms] AND (“Weight Loss”[MeSH Terms] OR “Nutritional Status”[MeSH Terms] OR “Cachexia”[MeSH Terms] OR “Malnutrition”[MeSH Terms] OR “Frailty”[MeSH Terms] OR “Sarcopenia”[MeSH Terms] OR “Muscular Atrophy”[MeSH Terms])) **NOT** (“case reports”[Publication Type] OR “comment”[Publication Type] OR “meta analysis”[Publication Type] OR “retracted publication”[Publication Type] OR “retraction of publication”[Publication Type] OR “systematic review”[Publication Type] OR “review”[Publication Type])
Embase	
#1	parkinson disease’/exp
#2	body weight loss’/exp OR ‘nutritional status’/exp OR ‘cachexia’/exp OR ‘malnutrition’/exp OR ‘frailty’/exp OR ‘sarcopenia’/exp OR ‘muscle atrophy’/exp
#3	(2000–2022)/py
#4	(‘editorial’/exp OR ‘editorial’ OR ‘review’/exp OR ‘review’ OR ‘letter’/exp OR ‘letter’ OR ‘case report’/exp OR ‘meta analysis’/exp)
(#1 AND #2 AND #3) NOT #4	(‘parkinson disease’/exp OR ‘parkinson disease’) AND (‘body weight loss’/exp OR ‘body weight loss’ OR ‘nutritional status’/exp OR ‘nutritional status’ OR ‘cachexia’/exp OR ‘cachexia’ OR ‘malnutrition’/exp OR ‘malnutrition’ OR ‘frailty’/exp OR ‘frailty’ OR ‘sarcopenia’/exp OR ‘sarcopenia’ OR ‘muscle atrophy’/exp OR ‘muscle atrophy’) **NOT** (‘editorial’/exp OR ‘editorial’ OR ‘review’/exp OR ‘review’ OR ‘letter’/exp OR ‘letter’ OR ‘case report’/exp OR ‘meta analysis’/exp) AND (2000–2022)/py
Cochrane	
#1	“Parkinson Disease”[MeSH Terms]
#2	“Weight Loss”[MeSH Terms] OR “Nutritional Status”[MeSH Terms] OR “Cachexia”[MeSH Terms] OR “Malnutrition”[MeSH Terms] OR “Frailty”[MeSH Terms] OR “Sarcopenia”[MeSH Terms] OR “Muscular Atrophy”[MeSH Terms]
#1 AND #2	(“Parkinson Disease”[MeSH Terms] AND (“Weight Loss”[MeSH Terms] OR “Nutritional Status”[MeSH Terms] OR “Cachexia”[MeSH Terms] OR “Malnutrition”[MeSH Terms] OR “Frailty”[MeSH Terms] OR “Sarcopenia”[MeSH Terms] OR “Muscular Atrophy”[MeSH Terms]))
Web of Science	
#1	TS = (‘parkinson disease’)
#2	((((((TS = (‘body weight loss’)) OR TS = (‘nutritional status’)) OR TS = (‘cachexia’)) OR TS = (‘malnutrition’)) OR TS = (‘frailty’)) OR TS = (‘sarcopenia’)) OR TS = (‘muscle atrophy’)
#3	PY = (2000–2022)
#4	DT = (Review)
(((#1) AND #2) AND #3) NOT #4	

**Table 2 nutrients-14-05194-t002:** PICOS structure’s criteria for study inclusion.

Population	Adults
Intervention	Screening tools: MNA, MNA-SF, SGA, PG-SGA, MUST, MEONF-II, SCREEN-II and Body Mass Index
Comparison	Not applicable
Outcome	Malnutrition, risk of malnutrition
Study design	Cross-sectional, case-control, cohort

**Table 3 nutrients-14-05194-t003:** Group characteristics.

Total Number of Subjects	5613
Men *n* (%)	3216 (57.3%)
Women *n* (%)	2296 (40.9%)
Not given	100 (1.8%)
Age	
Minimum age (yrs)	20
Maximum age (yrs)	96
Origin	
Asia	24
Europe	16
North America	4
South America	3
Africa	0
Australia	2

**Table 4 nutrients-14-05194-t004:** MUST results.

Author	Subgroups	Participants (*n*)	Hoehn & Yahr	MDS-UPDRS (Total)	Age ± SD (min, max)	Male *n* (%)	Female *n* (%)	Mean MUST Score ± SD	% Risk of Malnutrtion	% Malnourished	% Normal	% Medium Risk	% High Risk	% Low Risk
Barichella M. et al., 2013 [15]		208	2.3	13.7 (II), 23.2 (III)	67.8 ± 9.20 (32.88)	141 (67.8)	67 (322)	0.27 ± 0.54	17.2	5	77.8	Nd	Nd	Nd
Jaafar A.F. et al., 2010 [29]	North Tyneside	82	2.49	30.96	74.59 ± 8.87 (50.96)	34 (41.5)	48 (58.5)	Nd	Nd	Nd	Nd	15.4	8.1	76.5
	North Northumberland	53	2.6	36.96	75.33 ± 9.25 (53.93)	32 (46.3)	21 (53.7)	Nd	Nd	Nd	Nd			

Nd—no data; SD—standard deviation; II—part II of MDS-UPDRS; III—part III of MDS-UPDRS.

**Table 5 nutrients-14-05194-t005:** SGA results.

Author	Participants (*n*)	Hoehn & Yahr	MDS-UPDRS (Total)	Age ± SD (min, max)—Median	Male *n* (%)	Female *n* (%)	Median SGA Score ± SD	% SGA-B	% SGA-C	% risk of Malnutrition	% Malnutrition	% Normal
Sheard, J.M. et al., 2013 [30]	125	Nd	Nd	70 ± nd (35, 92)	73 (58.4)	52 (41.6)	3 ± nd	15	0	Nd	Nd	85
Sheard, J.M. et al., 2013 [31]	125	2 (median)	Nd	70 ± nd (35, 92)	74 (59.2)	51 (40.8)	3 ± nd	15.2	0	Nd	Nd	84.8

Nd—no data; SD—standard deviation.

**Table 6 nutrients-14-05194-t006:** PG-SGA results.

Author	Participants (*n*)	Hoehn & Yahr	MDS-UPDRS (Total)	Age ± SD (min, max)—Median	Male *n* (%)	Female *n* (%)	Median PG-SGA Score ± SD	% SGA-B	% SGA-C	% risk of Malnutrition	% Malnutrition	% Normal
Sheard, J.M. et al., 2013 [31]	125	2 (median)	Nd	70 ± nd (35, 92)	74 (59.2)	51 (40.8)	3 ± nd	15,2	0	Nd	Nd	84,8

Nd—no data; SD—standard deviation.

**Table 7 nutrients-14-05194-t007:** SCREEN-II and MEONF-II.

Author	Subgroups	Participants (*n*)	Hoehn & Yahr	MDS-UPDRS (Total)	Age ± SD (min, max)	Male *n* (%)	Female *n* (%)	SCREEN-II Score (Median) ± SD	MEONF-II Score (Median) ± SD	% risk of Malnutrition	% Malnutrition	% Normal	% Moderate Risk	% High Risk
Lindskov, S. et al., 2015 [16]	SCREEN-II baseline	65	Nd	Nd	68.1 ± 8.1 (48.9)	35 (53.8)	30 (46.2)	46 ± nd	Nd	Nd	Nd	8.1	22.6	69.4
	SCREEN-II follow up							46 ± nd	Nd	21	75.8	3.2	Nd	Nd
	MEONF-II baseline							Nd	0 ± nd	31	0	96.9	Nd	Nd
	MEONF-II follow up							Nd	0 ± nd		1.5	95.4		

Nd—no data; SD—standard deviation.

**Table 8 nutrients-14-05194-t008:** Nutritional status of patients using the MNA questionnaire.

Nutritional Status	Number of Patients	Percentage of Patients
Normal	1007	38%
At risk of malnutrition	634	23.9%
Malnourished	294	11.1%
Not given	713	27%

**Table 9 nutrients-14-05194-t009:** MNA questionnaire results.

Author	Subgroups	Participants (*n*)	Hoehn & Yahr	MDS-UPDRS (Total)	Age ± SD (min, max)	Male *n* (%)	Female *n* (%)	Mean MNA Score ± SD	% Risk of Malnutrition	% Malnourished	% Normal
Albay V.B. et al., 2020 [32]		75	2.5	Nd	66.84 ± 10.1 (nd)	42 (56)	33 (44)	23 ± 3.6	59	Nd	Nd
Yang, T. et al., 2020 [10]		556	2.41	25.02 (III)	68.37 ± 10.47 (36.92)	324 (58.3)	232 (41.7)	19.82 ± 2.18	30	39.2	30.8
Bazán-Rodríguez, L. et al., 2020 [33]	normal nutritional status	49	2.1	46.2	64.4 ± 12.3 (nd)	35 (71.4)	14 (28.6)	26.4 ± 1.5	34.3	8	56.3
	abnormal nutritional status	38	2.4	63.6	66 ± 13.9 (nd)	23 (60.5)	15 (39.5)	19.8 ± 3.2			
Pisciotta, M.S. et al., 2019 [34]		195	Nd	Nd	73.6 ± 7.2 (nd)	124 (64)	71 (36)	Nd	29	Nd	Nd
Paul, B. et al., 2019 [11]		75	Nd	Nd	63 ± 10.5 (30.80)	40 (53.3)	35 (46.6)	Nd	45.3	12	42.7
Tomic, S. et al., 2018 [35]		34	Nd	19.5 (III)	71.18 ± 7.2 (56.82)	19 (55.9)	15 (44.1)	22.1 ± 4.2	Nd	Nd	Nd
Tomic, S. et al., 2017 [36]		96	2	19.34 (III)	70.22 ± 8.6 (41.86)	57 (59.4)	39 (40.6)	22.14 ± 3.98	55.2	8.3	36.5
Ghazi, L. et al., 2015 [37]		143	2	Nd	61.44 ± 10.47 (35.91)	96 (67.1)	47 (32.9)	Nd	Nd	Nd	Nd
Fereshtehnejad S.M. et al., 2015 [17]	Early onset	45	2 (median)	34.3	61.3 ± 10.4 (20.77)	28 (62.2)	17 (37.8)	24.4 ± 4.3	Nd	Nd	Nd
	Typical onset	95	2 (median)	30.6		67 (70.5)	28 (29.5)	25.5 ± 2.8			
Fereshtehnejad, S.M. et al., 2014 [38]		150	2 (median)	31.7	60.8 ± 10.8 (32.84)	103 (68.7)	47 (31.3)	25.1 ± 3.3	25.3	2.1	72.6
Vikdahl M. et al., 2014 [21]		58	2 (median)	23.5 (III)	68.4 ± 8 (nd)	36 (62.1)	22 (37.9)	25.8 ± 2.3	14	0	86
Fereshtehnejad, S.M. et al., 2014 [12]		143	1.98	Nd	61.44 ± 10.47 (35, nd)	96 (67.1)	47 (32.9)	25.14 ± 3.37	18.2	3.5	78.3
van Steijn J. et al., 2013 [39]		102	2 (median)	Nd	76.4 ± 6 (65, nd)	54 (59.2)	48 (48.1)	26.5 ± nd	20.5	2	77.5
Wang, G. et al., 2010 [40]	MNA > 23.5	117	1.79	Nd	64.74 ± 9.61 (28.83)	75 (64.1)	42 (35.9)	Nd	19.66	1.71	78.63
	MNA ≤ 23.5		2.08		65.08 ± 8.9 (28.83)						
	total		2 (median)		64.81 ± 9.42 (28.83)						
Barichella M. et al., 2008 [18]	2004	61	Nd	Nd	70.5 ± 5.5 (65.87)	37 (59.2)	24 (40.8)	24.9 ± 1.6	22.9	Nd	77.1
	2007	35			72.5 ± 4.6 (68.86)	20 (57.1)	15 (42.9)	24 ± 2.5	Nd	Nd	Nd
Gruber, M.T. et al., 2020 [41]		92	3 (median)	Nd	73.6 ± 6.7 (56.83)	52 (56.5)	40 (43.5)	12 median ± nd	39.1	6.5	54.3
Lin, T.K. et al. [42]		82	Nd	Nd	67.4 ± 9.06 (nd)	58 (70.7)	24 (29.3)	24.78 ± 2.27	29.27	0	70.73
Bril A. et al., 2017 [43]		114	2	I 7.8, II 10, III 21.2, IV 5.4	66.1 ± 9.8 (nd)	60 (53)	54 (47)	24.7 ± 3.8	28.1	7	64.9
Fereshtehnejad S.M. et al., 2015 [44]		108	2	33.1	60.9 ± 10.7 (nd)	77 (71.3)	31 (28.7)	25.4 ± 3.7	Nd	Nd	Nd
Kim S.R. et al., 2014 [45]		102	2	Nd	61.2 ± 10.1 (31.81)	45 (44.1)	57 (55.9)	21.4 ± 6.2	26.5	25.5	48
Vikdahl M. et al., 2014 [19]		118	2	35.7	68.5 ± 9 (nd)	67 (56.8)	51 (43.2)	25.4 ± 2.4	Nd	Nd	Nd

Nd—no data; SD—standard deviation; I—part I of MD-UPDRS; II—part II of MDS-UPDRS; III—part III of MDS-UPDRS; IV—part IV of MDS-UPDRS.

**Table 10 nutrients-14-05194-t010:** Risk of malnutrition in study by Lindskov S. et al., 2015 [16].

Author	Subgroups	Participants (*n*)	Hoehn & Yahr	MDS-UPDRS (Total)	Age ± SD (min, max)	Male *n* (%)	Female *n* (%)	Mean MNA Score ± SD	% No Risk of Malnutrition	% High Risk of Malnutrition	% High Risk of Malnutrition
Lindskov S. et al., 2015 [16]	SCREEN-II baseline	65	Nd	Nd	68.1 ± 8.1 (48.9)	35 (53.8)	30 (46.2)	45 ± nd (median)	8.1	22.6	69.4
	SCREEN-II follow up							46 ± nd (median)	3.2	21	75.8
	MEONF-II baseline							0 ± nd	96.9	3.1	0
	MEONF-II follow up							0 ± nd	95.4	3.1	1.5

Nd—no data; SD—standard deviation.

**Table 11 nutrients-14-05194-t011:** MNA-SF results.

Author	Subgroups	Participants (*n*)	Hoehn & Yahr	MDS-UPDRS (Total)	Age ± SD (min, max)	Male *n* (%)	Female *n* (%)	Mean MNA-SF Score ± SD	% Risk of Malnutrition	% Malnourished	% Normal
Albay V.B. et al., 2020 [32]		75	2.5	Nd	66.84 ± 10.1 (nd)	42 (56)	33 (44)	11.4 ± 2.06	59	Nd	Nd
Umay E. et al., 2021 [22]		54	Nd	Nd	67.25 ± 3.32 (65.80)	35 (64.8)	19 (35.2)	Nd	14.8	5.6	Nd
Wakabayashi H. et al., 2016 [46]		18	Nd	Nd	81.27 ± nd (70.94)	3 (16.7)	15 (83.3)	6.11 ± nd	Nd	Nd	Nd

Nd—no data; SD—standard deviation.

**Table 12 nutrients-14-05194-t012:** BMI results.

Author	Subgroups	Participants (*n*)	Hoehn and Yahr	MDS-UPDRS (Total)	Age ± SD (min, max)	Male *n* (%)	Female *n* (%)	BMI (kg/m^2^) ± SD	% Risk of Malnutrition	% Malnourished	% Normal
Barichella M. et al., 2013 [15]		208	2.3	13.7 (II), 23.2 (III)	67.8 ± 9.2 (32.88)	141 (67.8)	67 (32.2)	26.8 ± 4.2	17.2	5	77.8
Jaafar A.F. et al., 2010 [29]	North Tyneside	82	2.3	30.96	74.59 ± 8.84 (50.96)	34 (41.5)	48 (58.5)	24.88 ± 4.8	Nd	Nd	Nd
	North Northumberland	53	2.49	36.96	75.33 ± 9.25 (53.93)	32 (46.3)	21 (53.7)	26.49 ± 5.2			
Albay V.B. et al., 2020 [32]		75	2.5	Nd	66.84 ± 10.1 (nd)	42 (56)	33 (44)	28.6 ± 4.13	51	Nd	41
Yang, T. et al., 2020 [10]		556	2.41	25.02 (III)	68.37 ± 10.47 (36.92)	324 (58.3)	232 (41.7)	23.07 ± 3.33	30	39.2	30.8
Bazán-Rodríguez, L. et al., 2020 [33]	normal nutritional status	49	2.1	46.2	64.4 ± 12.3 (nd)	35 (71.4)	14 (28.6)	28.1 ± 4.9	34.3	8	56.3
	abnormal nutritional status	38	2.4	63.6	66 ± 13.9 (nd)	23 (60.5)	15 (39.5)	25.8 ± 4.3			
Pisciotta, M.S. et al., 2019 [34]		195	Nd	Nd	73.6 ± 7.2 (nd)	124 (64)	71 (36)	Nd	29	0	71
Paul, B. et al., 2019 [11]		75	Nd	Nd	63 ± 10.5 (30.80)	40 (53.3)	35 (46.6)	Nd	45.3	12	42.7
Tomic, S. et al., 2018 [35]		34	Nd	19.5 (III)	71.18 ± 7.2 (56.82)	19 (55.9)	15 (44.1)	28.53 ± 4.9	Nd	Nd	Nd
Tomic, S. et al., 2017 [36]		96	2	19.34 (III)	70.22 ± 8.6 (41.86)	57 (59.4)	39 (40.6)	29.5 ± 5.9	55.2	8.3	36.5
Ghazi, L. et al., 2015 [37]		143	2	Nd	61.44 ± 10.47 (35.91)	96 (67.1)	47 (32.9)	25.86 ± 4.3	Nd	Nd	Nd
Fereshtehnejad S.M. et al., 2015 [17]	Early onset	45	2 (median)	34.3	61.3 ± 10.4 (20.77)	28 (62.2)	17 (37.8)	Nd	Nd	Nd	Nd
	Typical onset	95	2 (median)	30.6		67 (70.5)	28 (29.5)				
Fereshtehnejad, S.M.et al., 2014 [38]		150	2 (median)	31.7	60.8 ± 10.8 (32.84)	103 (68.7)	47 (31.3)	25.8 ± 4.2	25.3	2.1	72.6
Vikdahl M. et al., 2014 [21]		58	2 (median)	23.5 (III)	68.4 ± 8 (nd)	36 (62.1)	22 (37.9)	26.3 ± 3.8	14	0	86
Fereshtehnejad, S.M. et al., 2014 [12]		143	1.98	Nd	61.44 ± 10.47 (35, nd)	96 (67.1)	47 (32.9)	25.86 ± 4.3	18.2	3.5	78.3
van Steijn J. et al., 2013 [39]		102	2 (median)	Nd	76.4 ± 6 (65, nd)	54 (52.9)	48 (48.1)	25.2 ± 3.6	20.5	2	77.5
Wang, G. et al., 2010 [40]	MNA > 23.5	117	1.79	Nd	64.74 ± 9.61 (28.83)	75 (64.1)	42 (35.9)	Nd	19.66	1.71	78.63
	MNA ≤ 23.5		2.08		65.08 ± 8.9 (28.83)			23.54 ± 2.7			
	total		2 (median)		64.81 ± 9.42 (28.83)			20.06 ± 2.8			
Barichella M. et al., 2008 [18]	2004	61	Nd	Nd	70.5 ± 5.5 (65.87)	37 (59.2)	24 (40.8)	27.1 ± 3.1	22.9	Nd	77.1
	2007	35			72.5 ± 4.6 (68.86)	20 (57.1)	15 (42.9)	25.9 ± 3.5	Nd	Nd	Nd
Gruber, M.T. et al., 2020 [41]		92	3 (median)	Nd	73.6 ± 6.7 (56.83)	52 (56.5)	40 (43.5)	25.7 ± 3.4	39.1	6.5	54.3
Lin, T.K. et al., 2020 [42]		82	Nd	Nd	67.4 ± 9.06 (nd)	58 (70.7)	24 (29.3)	23.93 ± 3.1	29.27	0	70.73
Sheard, J.M. et al., 2013 [30]		125	Nd	Nd	70 ± nd (35.92) (median)	73 (58.4)	52 (41.6)	Nd	Nd	Nd	85
Sheard, J.M. et al., 2013 [31]		125	2 (median)	Nd	70 ± nd (35.92) (median)	74 (59.2)	51 (40.8)	26.1 ± nd (median)	Nd	Nd	84.8
Bril A et al., 2017 [43]		114	2	I 7.8, II 10, III 21.2, IV 5.4	66.1 ± 9.8 (nd)	60 (53)	54 (47)	29.4 ± 4.6	28.1	7	64.9
Petroni M.L. et al., 2003 [47]		35	Nd	Nd	69.7 ± 5.8 (nd)	20 (57.1)	15 (42.9)	25.3 ± 4.3	Nd	Nd	Nd
Salari M. et al., 2018 [48]		35	1.9	49.1	59.8 ± 11.4 (36.80)	25 (57.1)	10 (42.9)	24.8 ± 3.4	Nd	Nd	Nd
Ozdilek B. et al., 2014 [23]	UPDRS male	40	Nd	27.6	60.8 ± 9.4 (42.82)	28 (70)	12 (30)	30.17 ± 5.1	Nd	Nd	Nd
	UPDRS female		Nd	21.9							
Wilczynski J. et al., 2018 [49]		32	Nd	Nd	54.28 ± 12.24 (32.85)	6 (18.75)	26 (81.25)	24.12 ± 3.49	Nd	Nd	Nd
Tan Y.J. et al., 2020 [24]		102	I 1%, II 61.4%, III 23.8%, IV 8.9%, V 4%	36,1	68.2 ± 8.8 (41.87)	57 (55.9)	45 (44.1)	23.3 ± 3.7	Nd	Nd	Nd
Küsbeci Ö.Y. et al., 2019 [25]	Male	100	Nd	24.54 (III)	67.37 ± 8.47 (nd)	61 (61)	39 (39)	27.4 ± 3.58	Nd	Nd	Nd
	Female				63.35 ± 9.01 (nd)			28.89 ± 5.29			
Lopez-Botello C.K. et al., 2017 [50]	< 60 y.o.	15	I 27%; II 53%; III 20%	Nd	63.33 ± 9.4 (nd)	7 (47)	8 (53)	27.42 ± 5.28	Nd	Nd	Nd
	> 60 y.o.							27.75 ± 6.07			
Lindskov S. et al., 2015 [16]	BMI baseline	65	Nd	Nd	68.1 ± 8.1 (48.90)	35 (53.8)	30 (46.2)	27.7 ± 4.4	Nd	Nd	Nd
	BMI follow up							28 ± 4.8			
Cereda E. et al., 2012 [26]		80	Nd	Nd	61.5 ± 10.5 (39.84)	42 (52.5)	38 (47.5)	27.6 ± 5.1	Nd	Nd	Nd
Morales-Briceño H. et al., 2012 [51]		177	2.3	Nd	64.8 ± 12 (nd)	100 (56.5)	77 (43.5)	27.2 ± 4.7	Nd	Nd	Nd
Barichella M. et al., 2003 [52]		364	Nd	Nd	65.9 ± 8.9 (35.93)	184 (50.5)	180 (49.5)	27.3 ± 4.4	Nd	Nd	Nd
Tan A.H. et al., 2018 [53]		93	2,3	I 11.8, II 14.1, III 32.9, IV 3.4	66 ± 8.5 (38.84)	51 (54.8)	42 (45.2)	24 ± 0.4	Nd	Nd	Nd
Cheshire W.P. et al., 2004 [27]		100	Nd	Nd	74 ± 7.3 (59.89)	Nd	Nd	25.4 ± 4.1	Nd	Nd	Nd
Wu Q. et al., 2020 [13]	H&Y underweight	245	3.2	Nd	64.5 ± 10.9 (nd)	143 (58.4)	102 (41.6)	22.6 ± 3.4	Nd	Nd	Nd
	H&Y not underweight		2.4	Nd							
	UPDRS motor underweight		Nd	23.7 (III)							
	UPDRS motor not underweight		Nd	17.1 (III)							
Yoo HS. et al., 2019 [20]	Underweight	70	Nd	17.05 (III)	66.05 ± 6.79 (nd)	39 (55.7)	31 (44.3)	21.22 ± 1.82	Nd	Nd	Nd
	Normal weight			17.05 (III)	66.05 ± 6.79 (nd)			21.22 ± 1.82			
	Overweight			15.73 (III)	64.84 ± 6.48 (nd)			23.99 ± 0.72			
	Obese			18.48 (III)	64.94 ± 8.29 (nd)			27.21 ± 2.25			
Wu Q. et al., 2020 [28]	Male	253	2.6	18.9 (III)	63.8 ± 10.7 (nd)	146 (57.7)	107 (42.3)	22.2 ± 3.3	Nd	Nd	Nd
	Female		2.5	17.2 (III)	62 ± 9.9 (nd)			23.2 ± 3.5			
Yong, V.W. et al., 2020 [54]	Baseline	77	I 1.3%, II 64.5%, III 32.9%, IV 0%, V 1.3%	32,9	65.6 ± 8.9 (38.84)	43 (55.8)	34 (44.2)	24.3 ± 3.9	Nd	Nd	Nd
	Follow up		I 1.3%, II 52.6%, III 30.3%, IV 9.2%, V 5.3%	39.3				23.1 ± 4			
Yazar T. et al., 2018 [55]	Male	166	Nd	Nd	72.76 ± 4.42 (nd)	83 (50)	83 (50)	26.41 ± 3.38	Nd	Nd	Nd
	Female				71.57 ± 5.2 (nd)			26.45 ± 4.6			
Zilli Canedo Silva M. et al., 2015 [56]		17	I 47%, II 47%, III 6%	Nd	58.9 ± 12.8 (nd)	9 (52.9)	8 (47.1)	26.3 ± nd (median)	Nd	Nd	Nd
Fereshtehnejad S.M. et al., 2015 [44]		108	2	33.1	60.9 ± 10.7 (nd)	77 (71.3)	31 (28.7)	Nd	Nd	Nd	Nd
Lee, C.Y. et al., 2019 [57]	No sarcopenia	31	1.6	28.2	61.7 ± 10.6 (nd)	21 (40.4)	31 (59.6)	26.1 ± 3.1	Nd	Nd	Nd
	Sarcopenia	21	1.6	42.7				21.4 ± 3.5			
Peball M. et al., 2018 [14]	Total	104	2.5	I 12.7, II 17.1, III 38.4, IV 4.1	73.8 ± 5.2 (nd)	64 (61.5)	40 (38.5)	25.1 ± 3.6	Nd	Nd	Nd
	No sarcopenia	46	1.9	Nd	73.3 ± 5.7 (nd)			25 ± 3.3			
	Sarcopenia	58	2.9	Nd	74.2 ± 4.8 (nd)			25.2 ± 3.9			
da Luz, M.C.L. et al., 2020 [58]	No sarcopenia	77	Nd	36.5	65.4 ± 8.9 (nd)	45 (58.4)	32 (41.6)	27.7 ± 4.5	Nd	Nd	Nd
	Sarcopenia			41.3				24.8 ± 1.9			
Kim S.R. et al., 2014 [45]		102	2	Nd	61,2 ± 10.1 (31.81)	45 (44.1)	57 (55.9)	23.2 ± 3.7	26.5	25.5	48
Umay E. et al., 2021 [22]		54	Nd	Nd	67.25 ± 3.32 (65.80)	35 (64.8)	19 (35.2)	26.55 ± 3.69	14.8	5.6	79.6
Vikdahl M. et al., 2014 [19]		118	2	35.7	68.5 ± 9 (nd)	67 (56.8)	51 (43.2)	26.3 ± 3.9	Nd	Nd	Nd
Wakabayashi H. et al., 2016 [46]		18	Nd	Nd	81.27 ± nd (70.94)	3 (16.7)	15 (83.3)	17.84 ± nd	Nd	Nd	Nd

Nd—no data; SD—standard deviation; I—part I of MD-UPDRS; II—part II of MDS-UPDRS; III—part III of MDS-UPDRS; IV—part IV of MDS-UPDRS; I%—a percentage of patients at stage I determined by Hoehn & Yahr scale, II%—a percentage of patients at stage II determined by Hoehn & Yahr scale, III%—a percentage of patients at stage III determined by Hoehn & Yahr scale, IV%—a percentage of patients at stage IV determined by Hoehn & Yahr scale.

## Data Availability

Not applicable.

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
