# Peer review of "Prevalence of Malnutrition in Patients with Parkinson’s Disease: A Systematic Review"

_nutrients, 2022, doi:10.3390/nu14235194_

Round 1

Reviewer 1 Report

This study reviews the data about nutritional status and malnutrition in patients with Parkinson Disease. This study has gathered a punch of interesting researchers and has shed light to an important issue that may affect seriously in the daily life of the people and that sometimes may be undervalued. Nevertheless, there are several areas needed to be considered before publication. Here are my key concerns and they are not in order of importance:

-        Was this systematic review protocol registered in the Prospective International Register of Systematic Reviews, PROSPERO?

-        The font varies in the text.

-        Use the PRISMA template.

-        The quality assessment figures could go in annexes since there are several of them and they are large figures.

-        The authors use several databases (Cochrane, PubMed, Embase and Web of Science) and I would recommend differentiating them in the flow diagram, as it appears in the PRISMA guidelines.

-        In introduction and discussion parts, it is highly recommended a second review, some paragraphs lack coherence as some sentences are too short and very little connectors or linkers are used (e.g. some sentences could go together, thus giving more clarity). Plus, some words are misspelled (minor error).

-        The introduction section is quite long. The list of previous evidences could be summarized and reorganized to better highlight what this review adds to the existing literature.

-        Change the tables of each information about the measurements from results to data extraction section.

-        In order to enrich your results reflection in the discussion part, it is suggested to give some advice to improve the study of malnutrition and its adequacy (e.g. what it is needed to change / specify, which could be the predominant factors to consider when designing such scale/assessment).

Author Response

Dear Sirs,

Thank you for giving us the opportunity to submit a revised draft of our manuscript titled „Prevalence of malnutrition in patients with Parkinson’s Disease: a systematic review”.

We appreciate the time and effort that you and the Reviewers have dedicated in order to provide valuable feedback on our manuscript. We have been able to incorporate changes and reflect on all Reviewer’s suggestions. We have highlighted the changes within the manuscript in red. Here is a point-by-point response to the Reviewers’ comments and concerns.

Comments from Reviewer 1.

Dear Reviewer,

Thank you for your comments, they were very important in improving our review. Our review was not registered in the Prospective International Register of Systematic Reviews, PROSPERO. We have checked the font in the article one more time and changed all texts to be written with the same font. We also systemized again step by step the PRISMA protocol sections. Unfortunately, some parts of the protocol were dedicated to meta-analysis and we were not able to apply them due to the lack of possibility to synthesize data. In accordance to your suggestion, we followed other systematic reviews published by Nutrients.

Based on our best knowledge, the quality assessment figures could go in the annexes but it’s up to the Editor before the publication, it is not our choice.

We have also added more detailed information on the research in databases to the flowchart, as you have wisely suggested.

We have consulted other authors about moving the tables about the measurements from the results to the data extraction section and decided to leave it in the results section since those tables can be placed in both sections and each way is correct.

We added in the introduction additional information highlighting the importance of routine screening for malnutrition in patients with Parkinson’s disease. We made it a bit shorter and reorganized it for better insight into the problem of malnutrition.

We have also extended the discussion enriching it with more detailed information about available data and suggested changes be made in the evaluation of every patient diagnosed with Parkinson’s disease according to the worldwide recommendations by most important societies.

We hope that our paper improvements will fulfill your expectations. Again, thank you for your time, we appreciate it a lot because thanks to you we can improve and learn more.

Kindest regards,

Authors

Reviewer 2 Report

Kacprzyk and colleagues compiled 49 studies to analyze the prevalence of malnutrition among patients with Parkinson's disease. The authors wrote this review with PRISMA guidelines and extensively searched the literature database.  They assessed the quality of each study and compared results in each nutritional status assessment subcategory. The authors summarized the difficulties in the cross-study comparison of using different assessment tools (questionnaires) without offering sufficient introduction or comparisons between the questionnaires. More importantly, the authors failed to reach conclusions based on the comparison or consolidate any current knowledge or prospects. 

Specific points:

1. The authors introduced the risk of malnutrition in PD patients with limited background information and insufficient mechanisms.  Only one condition associated with malnutrition in PD, oropharyngeal dysphagia, was discussed.  The basic mechanisms behind malnutrition in light of energy homeostasis were omitted in this section. In addition, other postulated determinants of weight loss in PD, such as hyposmia, cognitive impairment, and gastrointestinal dysfunction, were insufficiently discussed in the manuscript. 

2. The PD pathology and treatment were presented with minimal references in the introduction. Only three articles were selected without any in-depth discussion. The occurrence of weight loss in PD and its clinical implication was not reported in detail. Moreover, the authors did not present any possible clinical benefit of addressing malnutrition in PD or therapeutic alternative in the future management of PD regarding prevalent malnutrition. 

3. The comparison in each subcategory of nutritional status assessment was informative. However, the authors did not introduce or compare each assessment tool. The scope and parameters of each questionnaire need to be compared with discussion and necessary references. 

4. Other nutritional assessment measures, such as biochemical and anthropometric measurements other than BMI, were not presented or discussed in the paper. Those measurements, especially biochemical indicators, have universal standards and are more convenient to track for longitudinal changes. 

5. The authors failed to reach any applicable conclusions. Even though the cross-comparison is challenging, the authors should at least compare this review with the previous systematic review on the same subject to update any current knowledge or prospects. 

Author Response

Dear Sirs,

Thank you for giving us the opportunity to submit a revised draft of our manuscript titled „Prevalence of malnutrition in patients with Parkinson’s Disease: a systematic review”.

We appreciate the time and effort that you and the Reviewers have dedicated in order to provide valuable feedback on our manuscript. We have been able to incorporate changes and reflect on all Reviewer’s suggestions. We have highlighted the changes within the manuscript in red. Here is a point-by-point response to the Reviewers’ comments and concerns

Comments from Reviewer 2.

Dear Reviewer,

Thank you for your comments, they were highly important in improving our review. Your comment about introducing every questionnaire was very on point, we put detailed information about each screening tool in the discussion section and also compared it. We also fulfilled your suggestions about extending the part about pathology, treatment and aspects such as dysphagia, weight loss, gastrointestinal dysfunction, hyposmia or cognitive impairment. We highlighted the importance of nutritional status screening as you have wisely suggested and also added a few recommendations for both caretakers and doctors taking care of PD patients. We included GLIM criteria and ESPEN guidelines in clinical nutrition in neurology to show even more how important the screening is. We summarised the most current knowledge in the discussion section of our manuscript.

We have omitted some other tools used for nutritional measurements such as biochemical markers and anthropometric measurements because our paper was based solely on screening assessment done via questionnaires since those are important as well. Biochemical markers play an important role in the assessment of a patient’s nutritional status and could be used as well but we focused more on screening tools to show how restricted those can be and how important it is to do routine screening in all patients. The next stage of our work will be a more detailed analysis of the problem.

We have highlighted the importance of routine screening that should be applied to every patient diagnosed with Parkinson’s disease and later throughout the course of the disease. We also pointed out the need for assessing dysphagia and swallowing dysfunctions as it is strongly associated with malnutrition in this group.

We hope our paper improvements will fulfill your expectations. Thank you one more time for your time, we appreciate it a lot since we can improve and learn a lot.

Kindest regards,

Authors

Reviewer 3 Report

Dear Authors,

In their review, they address the prevalence of malnutrition in patients with Parkinson's disease.

A systematic review of the current state of the literature can be interesting, but only if a very specific question is addressed and then discussed intensively. Neither is the case in the present paper.

Instead of repeating rather superficial phrases about BMI over and over again (and also without discussing them critically), it would be more purposeful, for example, to look for differences in the course of the disease (country, region, age, sex, etc.), which could possibly be associated with different diets.

Since neither a clearly focused question nor a critical discussion with possible solutions is recognizable, I recommend rejection.

Author Response

Dear Sirs,

Thank you for giving us the opportunity to submit a revised draft of our manuscript titled „Prevalence of malnutrition in patients with Parkinson’s Disease: a systematic review”.

We appreciate the time and effort that you and the Reviewers have dedicated in order to provide valuable feedback on our manuscript. We have been able to incorporate changes and reflect on all Reviewer’s suggestions. We have highlighted the changes within the manuscript in green. Here is a point-by-point response to the Reviewers’ comments and concerns.

Comments from Reviewer 3.

Dear Reviewer,

Thank you for your short comments on our paper. We have applied all the comments other Reviewers provided for us and we hope our paper has improved.

To address your comment about looking into differences in the course of the disease, we have mentioned in the conclusions that aspects, such as sex, diet, age, and severity of the disease can have an impact as well on the prevalence of the disease and we hope to look deeper into what you have suggested in our next paper with more detailed analysis.

We hope our improved paper makes you more interested. Thank you one more time for your time, we appreciate it a lot.

Kindest regards,

Authors

Round 2

Reviewer 2 Report

The authors have made significant changes to the introduction and discussion of the manuscript following the suggestions in the previous review. Adding a detailed comparison of questionnaires was helpful to the readers. The clarity and soundness of the paper have been improved. However, there are more concerns to address before publication:

1. The authors should refer to the original articles instead of copying the clinical nutritional guideline (reference 4) when introducing the side effects of antiparkinsonian drugs in lines 51-53 or mass gain post-DBS treatments in lines 67-69. 

2. Although the authors added one sentence in lines 276-278, they need to elaborate on energy homeostasis as the primary mechanism behind malnutrition in PD, including the possible ways PD symptoms result in higher energy expenditure. 

3. In lines 288-289, the authors stated that early detection of weight loss could help reduce disease progression. However, no references or studies were listed to support this conclusion. The authors did not expand on the clinical benefit of addressing malnutrition in PD or therapeutic alternatives in PD management with enough clarity or examples. 

4. The manuscript still lacks a clear conclusion. With all the numeric tables and 49 records, the authors need to provide a concise and robust message for the readers to take home after reading the paper. 

Author Response

Dear Reviewer,

Thank you for your comments, we have applied all changes you have suggested. We have changed the citations when introducing side effects of antiparkinsonian drugs or mass-gain post-DBS treatments into the original articles. We have also included more information on energy homeostasis as the primary mechanism behind malnutrition in PD patients. We have also added clinical benefits from addressing malnutrition and early detection in weight loss, as you have suggested. We also changed the conclusion section in order to provide a more clear conclusion.

We hope that our paper improvements will fulfill your expectations. Again, thank you for your time, we appreciate it a lot.

Kindest regards,

Authors

Reviewer 3 Report

Dear authors,

Unfortunately, I do not find this new version of the paper much better. In the discussion, you have presented the differences between the various questionnaires and also briefly touched on the factors that play a role in PD, but you have not taken this discussion to its logical conclusion, e.g., by giving recommendations on how the questionnaires should be adapted. 

Critically, I find sentences like: L 373 Other important factors that can provide more detailed insight into the problem of malnutrition are biochemical markers.

These markers are also mentioned and then the whole thing concludes with the sentence: but it was not included in our study.

The general problem of the review is that it lacks a focus on a specific subgroup of PD patients e.g. in terms of gender and interaction of malnutrition. The overall discussion is a bit better, but the bottom line of the paper "malnutrition" is important in PD, does not tell the reader how to do it better. Neither information about which questionnaire should be used in PD patients now, nor for what reasons which information should be collected in addition in the future is given.

And then in the Conclusion comes a sentence: L 433 It was not possible to assess differences between sex, diets, region, economic status, or the severity of Parkinson's disease but it can provide even more insight and help to differentiate more aspects that determine a patient "at risk".

1. in the conclusion new factors are added, which were not addressed and discussed before. Are these important or not?

2. which "even more insight" have I got now? And which aspects do I differentiate now?

So I need more and better questionnaires. --> But what makes my questionnaire better?

I don't see any added value of this review.

Author Response

Dear Reviwer,

Thank you for your opinion and time, we appreciate it a lot. Your remarks are of great value to us.

Kindest regards,

Authors